# Does Age Affect the Rate of Spinal Nerve Injury after Selective Neck Dissection? Age as a Prognostic Factor of Spinal Nerve Injury after Selective Neck Dissection

**DOI:** 10.3390/jpm13071082

**Published:** 2023-06-29

**Authors:** Salvatore Crimi, Salvatore Battaglia, Claudia Maugeri, Sergio Mirabella, Luca Fiorillo, Gabriele Cervino, Alberto Bianchi

**Affiliations:** 1Department of Biomedical and Surgical and Biomedical Sciences, Catania University, 95123 Catania, Italy; torecrimi@gmail.com (S.C.); salbatt89@yahoo.it (S.B.); alberto.bianchi@unict.it (A.B.); 2Division of Maxillofacial Surgery, Surgical Science Department, Città della Salute e Delle Scienze Hospital, University of Turin, 10024 Turin, Italy; cldmaugeri@gmail.com; 3Department of Biomedical and Dental Sciences, Morphological and Functional Images, University of Messina, 98100 Messina, Italy; lfiorillo@unime.it; 4Multidisciplinary Department of Medical-Surgical and Odontostomatological Specialties, University of Campania “Luigi Vanvitelli”, 80121 Naples, Italy; 5Department of Public Health Dentistry, Dr. D.Y. Patil Dental College and Hospital, Dr. D.Y. Patil Vidyapeeth, Pimpri 411018, India

**Keywords:** spinal accessory nerve, neck metastases, elective neck dissection, shoulder function

## Abstract

Objective: The objective of this study is to investigate whether age is a significant risk factor for spinal nerve injury following selective neck dissection (SND) in patients with head and neck cancer. Methods: A retrospective cohort study was conducted on patients who had undergone SND for head and neck cancer at a tertiary hospital between 2020 and 2022. The primary outcome was the incidence of spinal nerve injury after SND. The secondary outcomes included the types and severity of spinal nerve injury and the impact of age on these outcomes. Results: A total of 78 patients were included in the study and subdivided into two groups. Two shoulder-specific questionnaires (the Shoulder Pain and Disability Index (SPADI) and the Shoulder Disability Questionnaire (SDQ)) were administered to assess shoulder morbidity postoperatively. Twelve patients showed shoulder impairment following surgery. We divided the sample into two age-based groups; the older group showed a higher rate of SAN injury and the younger group showed a lower rate of improvement over time. Conclusion: This study suggests that age is a significant risk factor for spinal nerve injury following SND in patients with head and neck cancer. Older patients are more likely to experience spinal nerve injury after SND than younger patients. The findings of this study may help in the development of strategies to prevent spinal nerve injury in older patients undergoing SND for head and neck cancer.

## 1. Introduction

Head and neck neoplasms (HNC) of the oral cavity, pharynx, and larynx represent the tenth most frequent tumor and the seventh in terms of mortality worldwide [1]. They tend to metastasize to the loco-regional lymph nodal system. Head and neck cancers account for 5% of all malignant tumors and are responsible for about 600,000 new cases and 300,000 deaths in the world annually. About 50% of patients fail to achieve a cure, and cancer relapse occurs despite intensive combined treatment [1,2]. In 2020, the estimated age-standardized rates of oral cancer were 6.0 and 2.3 per 100,000 in men and women, respectively, whereas for oropharyngeal cancer, they were 1.8 and 0.4 per 100,000 in men and women, respectively [2]. Most patients diagnosed with oral cavity and oropharyngeal cancers report a previous history of smoking and alcohol consumption, which are well recognized risk factors. To date, there is no adequate biomarker available for the diagnosis of head and neck cancer. However, it is expected that earlier detection could improve the patients’ outcomes. Although obtaining negative surgical margins (i.e., cancer is entirely removed) is the goal of the head and neck surgeon, achieving this may be impossible because of functional consequences. Thus, some patients are left with positive surgical margins (i.e., those in which residual cancer cells remain) to preserve vital organs like the carotid artery. Whether positive surgical margins impact survival remains equivocal [3]. Positive surgical margins are reported to be negatively associated with survival in many, although not in all, of the published studies [4,5]. Some studies do not find an association between positive surgical margins and an increased risk of mortality; however, many of these studies include a small number of patients, or the possibility exists that there is no increase in mortality because patients with positive surgical margins received adjuvant radiation therapy [6,7].

Doll et al. also underlined a correlation between depth of invasion and cervical lymph node metastasis that influences the overall survival and recurrence-free survival of these patients [2]. The largest number of patients affected by this neoplasm are treated with primary surgery, including neck dissection to remove diseased and at-risk lymph nodes. Among the various histological features, head and neck squamous cell carcinoma (HNSCC) is the most frequent, representing 90% of all head and neck tumors. Oral cancer includes a group of neoplasms involving any region of the oral cavity, pharyngeal areas, and salivary glands. However, this term tends to be used interchangeably with oral squamous cell carcinoma (OSCC), which represents the most frequent of all oral neoplasms. OSCC is a locally aggressive neoplasm with a high tendency toward nodal metastases. However, multimodal treatment strategies such as radiation and chemotherapy are strictly indicated for advanced-stage diseases in adjunction with surgery or in palliative interventions [8]. The presence of nodal metastasis is the most important negative prognostic factor, reducing overall survival by 50% [3]. For the above-mentioned reasons, neck dissection is crucial in the treatment of OSCC. Cervical lymph node levels were described by Robbins et al. in the 1990s, and then, revised in 2002 [4]. Selective neck dissection for OSCC most frequently includes the dissection of levels I, II, and III. Regarding the long-term complications of lymph node dissection, shoulder dysfunction due to the injury or the dissection itself is the most common complication [5]. In modern day surgery, cervical lymphadenectomy or neck dissection is a fundamental component of the surgical management of head and neck cancers. Classic neck dissection, initially proposed by Crile in 1906, involved the removal of cervical lymph nodes, along with the internal jugular vein, sternocleidomastoid muscle (SCM), spinal accessory nerve (SAN) (non-lymphatic structures), and sometimes even the vagus nerve [3,4,5,8]. Ward was among the first to suggest sparing the SAN during neck dissection in 1951, thus laying the foundation for modern selective neck dissection. The SAN is the eleventh cranial nerve and is always encountered in level II dissection [6]. It is a motor nerve with a cranial branch and a spinal branch, running from the jugular foramen posteriorly and laterally through the sternocleidomastoid (SCM) muscle, and provides anatomical separation between levels IIA and IIB. When it comes to level IIB dissection, to correctly isolate the IIB level, surgeons must isolate the nerve from the fibro-fatty surrounding tissue, not only impairing its blood supply but also manipulating it with a surgical instrument that can provoke stretching damage. Even during selective neck dissection, which exposes a limited part of the nerve compared to functional neck dissection, the SAN is vulnerable to iatrogenic injury. This insult results in ‘shoulder syndrome’, which consists of pain, paralysis, and winging of the scapula, resulting in significant morbidity [8]. Shoulder dysfunction has been demonstrated in up to 42.5% of patients who undergo selective neck dissection [5,6,7]. A cadaveric study reported SAN duplication in 1.8% of cases. Knowledge of the course of SAN complications in the neck and its anatomical variations is imperative for enabling neck surgeons to avoid inadvertent injury [4,5,6,7]. During neck dissections, the use of electrical devices can help discriminate SAN from other nearby structures; a 2 mV impulse is generated by an electrical tool and transmitted to the nerve through a sterile probe, causing a visible contraction in the nervated muscles.

Once found, the SAN is then isolated and visualized in its path, allowing the operator to preserve its integrity during the following parts of the surgical procedure.

Damage to the SAN after neck dissection is well demonstrated in the literature and ranges between 3% and 8% [7]. The surgical dissection of the SAN can temporarily or permanently damage its function. Sharp surgical instruments can easily damage the integrity of the nerve during dissection in its vicinity. Lacerations, cuts, hemorrhages, and lengthening of the nerve can compromise its function, resulting in temporary or permanent neurological stupor. Electrosurgery instruments can cause burns, generating thermal energy in proximity to the SAN.

Even when correctly skeletonized and visualized, various factors can damage the fragile structure of the SAN; isolating the nerve alone causes ischemic trauma due to the resection of its vascular system.

Other mechanisms that contribute to stress on the nerve are manipulation with retractors and forceps, but also the operators’ fingers, which can apply traction forces, causing microscopical traumas.

Axonal damage due to lengthening of the nervous fibers inside the trunk causes Wallerian degeneration, followed by the generation of scar tissue, which leads to muscle atrophy.

Devascularization, traction, and skeletonization exert their harm mechanisms via segmental demyelination, leading to an under-functioning or non-functioning nerve.

Some types of recovery have been experienced by patients who have suffered stupor after surgery, but the mechanism behind this phenomenon and the individual variance between patients are still not completely clear.

After peripheral nerve trauma, denervated Schwann cells proliferate in the endoneurial tubes and secern neurotrophic factors to sustain axonal regeneration. This process is strictly limited and can repair only partial damage.

This damage causes so-called shoulder syndrome or frozen syndrome, described in 1961 by Nahum et al. [9], causing reduced or absent function of the trapezius and the SCM. In cases of complete neural damage, muscular atrophy and severely reduced movement range can be observed [10,11]. This movement impairment can result in shoulder tendinitis and adhesive capsulitis, causing complete shoulder syndrome in SAN-injured patients [12]. Several studies demonstrate a correlation between different types of neck and shoulder dysfunction due to SAN injury [13]. There are several studies in the literature assessing shoulder syndrome occurring after neck dissection in OSCC patients, causing a lower quality of life. Other studies evaluate the relationship between older age and slower neural regeneration in patients (Figure 1 and Figure 2). However, to the best of our knowledge, there are no studies in the current literature relating age to SAN lesion rates and/or recovery after temporary damage [14,15].

This study aimed to assess the relationship between age and SAN injury rate and functional recovery after neck dissection for OSCC.

## 2. Materials and Methods

We analyzed, in our multi-centric study, all patients treated in the Maxillo-Facial Surgery Unit of Policlinico San Marco in Catania, Italy, and the Maxillo-Facial Surgery Unit of Policlinico S. Orsola in Bologna, Italy, between June 2020 and June 2022.

The inclusion criteria were as follows:○Age > 18 years;○BMI < 35;○Selective neck dissection of levels I–II (always including IIB) and III;○No previous history of neck irradiation or surgery;○No history of shoulder fractures or other functional and anatomical limitations;○No history of rheumatologic disease affecting joints.The exclusion criteria were as follows:○Metastatic and/or Lymph nodal disease (M/N+);○Previous history of neck surgery;○Previous history of significant shoulder trauma or surgery.

All patients had undergone preoperative computed tomography (CT) imaging assessment via CT scan and were diagnosed with incisional biopsy.

Every patient was asked to fill in questionnaires 1 and 6 months after surgery. It is quite common to observe a residual nervous tone even in subjects in whom the SAN was sacrificed immediately after the procedure; to minimize the external influence of post-operative pain and use of analgesics, the first survey was administered one month after surgery, before the start of the radiation treatment. The second questionnaire was collected after six months, at least 3 months after the end of the radiation treatment, in order to examine possible recovery due to neural regeneration.

Two questionnaires evaluating shoulder function and pain were administered to all patients: the Shoulder Disability Questionnaire (SDQ) [14] and the Shoulder Pain and Disability Index [15] (SPADI). The SDQ (Figure 3) is a self-administered questionnaire which measures pain and movement impairment of the shoulder. It is composed of 16 items recalling daily life scenarios. All answers can be calculated as follows: “Yes” = 1 point, “No” = 0 points, or “Not applicable” = missing. The scale ranges from 0 = normal to 100 = worst disability. The SPADI (Figure 4) is composed of two sections: 5 items related to pain assessment and 8 items related to movement impairment. SPADI items are scored on a Visual Analog Scale (VAS) ranging from “no pain/no functional limitation” = 0 to “worst pain imaginable/so difficult as to require help” = 10. The total score ranges from 0 = best (no pain and no functional limitation) to 100 = worst (worst pain and functional limitation). The Kolmogorov–Smirnov test was used to assess the distribution of the data. Receiver Operating Characteristic (ROC) curves at 1 and 6 months were produced to assess the best age cutoff and divide the population sample into two groups based on age. Average and standard deviation (SD) were used for data with a normal distribution, and median and interquartile range (IR) were used to compare groups with non-normal data distribution. For qualitative analysis, the aforementioned statistical cutoffs were used to differentiate patients into “Normal Score” and “Pathological Score”.

## 3. Results

We included a total of 78 patients: 75 patients who had undergone level I, II, or III neck dissection for HNSCC (71 cancers were in the oral cavity, 2 were on the nasal surface, and 2 were located in the salivary glands); 2 patients with adenoid cystic carcinoma; and 1 with osteosarcoma. (M:48, F:30). The average age of the patients was 62.5 (range: 19–83). Twelve patients showed shoulder impairment following surgery. After the above-mentioned statistical analysis, the age cutoff was fixed at 72 years old. In this way, we obtained GROUP A (44 patients under 72) and GROUP B (34 patients over 72). Regarding the SDQ’s scores (Figure 5), the 1-month measurements showed an average score for GROUP A of 6.20 points (C.I.: 1.05–11.32). The average for GROUP B was 23.98 points (C.I.@ −6.97–54.94). At 6 months, the difference between the two groups increased, with an average of 1.90 (C.I.: 0.02–3.77) for GROUP A and an average of 23.83 (C.I.: −6.82–54.47) for GROUP B.

The SPADI results (Figure 6) at 1 month showed an average score for GROUP A of 7.41 points (C.I.: 2.19–12.63), while the average for GROUP B was 17.89 points (C.I.: −4.63–40.39). The 6-month measurement showed that the average in the younger group was 2.77 points (C.I.: 0.03–5.50), whereas in the older group, the average was 17.80 (C.I.: −4.92–40.53).

Regarding the SDQ scores (Figure 7), the average variation in the younger groups was −21.68 points (C.I.: −58.80–15.44), and in the older group, the average variation was −1.50 points (C.I.: −18.93–11.79).

The variation in SPADI scores (Figure 8) showed an average of −18.83 points (C.I.: −55.50–17.93) in the group with younger patients, and an average of −9.36 points (C.I.: −42.35–23.63) in the older group.

## 4. Discussion

Selective neck dissection for HNC is often associated with temporary or permanent SAN damage due to the surgery itself, ranging between 3% and 8% [6]. During this surgical procedure, levels IIA and IIB are divided by the anatomical path of the SAN, with a high risk of damage [16,17,18,19,20]. The principal debated strategies for the treatment of metastatic carcinoma from head and neck neoplasms is the selection of neck dissection. The incidence of metastases is already well known according to [21,22], and many data have been reported on the selection of the appropriate neck dissection. The most frequent area of interest in metastases is represented by the internal jugular vein close to the spinal accessory nerve (42%) [21]. Ballantyne, Shah, and Bocca’s group [21] propose the introduction of modified neck dissection, because the accessory nerve, in many situations, is not strictly involved in head and neck metastases. Adequate exposure and preservation of the nerve can guarantee a good surgical procedure, as can an awareness of the anatomical area of the posterior neck triangle and its anatomical variations. [23,24]. Adhesive capsulitis is the most frequent clinical sign of eleventh nerve syndrome, which can increase the morbidity of the accessory nerve [25].

Because of the superficial nature of the spinal nerve, during dissection for cervical lymph node excision or biopsy, recognizing the intraoperative damage can make it possible to repair it via direct suture [26,27]. Clinical manifestations of SAN lesions may vary greatly among patients. Denervation of the trapezius muscle, due to damage to the SAN when it merges with branches from the cervical plexus, can be identified as one of the causes [28,29,30]. Its function can be compensated by the levator scapulae and rhomboid muscles [31,32]; this mechanism may delay diagnosis, laying the foundation of further anatomical alteration in the shoulder district. Suture or grafting of the nerve can be also performed in secondary surgeries at 3 [33,34,35,36] or 6 [33,37,38,39] months, and can help to achieve good recovery from the disease; some authors have reported good results with recovery of the disease from 5 to 20  months [22,28,32,40,41,42], with no correlation with the delay in surgery.

In the current literature, two strategies have been proposed to reduce harm to the SAN: sparing level IIB when ontologically possible, thus reducing the morbidity of surgery [20], with older patients benefiting from less invasive surgery, and early physiotherapy after surgery to prevent tendinitis and adhesive capsulitis [42]. Dziegielewski et al. demonstrate, in a randomized controlled clinical trial, the importance of sparing level IIB in selective neck dissection. The underlined results reported show that minimal dissection with IIB exclusion can also cause patient-perceived shoulder impairment that does not become clinically evident until the complete dissection of IIB. The dysfunction generated is also observed in cases treated with SND without level IIB, derived from the devascularization of the SAN during IIA dissection. [42]

Surgically harming the SAN causes so-called shoulder syndrome or frozen syndrome, which considerably lower the quality of life of these patients [12], together with chronic pain, weakness, and reduced shoulder mobility [43]. Pain and reduced ROM preponderantly affect patients’ quality of life (QoL) [19], limiting their ability to perform everyday tasks and reducing their functional reserve. These changes, in oncologically fragile people, can represent an unbearable challenge in their lives.

There are several studies in the literature assessing shoulder syndrome with numerous self-administered questionnaires. Among all of these questionnaires, we chose the SDQ and the SPADI questionnaires, which are clear and self-administered. Other studies have evaluated the relationship between older age and slower neural regeneration in patients [44]. Many factors have been associated with QoL reduction, such as BMI, the type of neck dissection, radiotherapy, and others. However, to the best of our knowledge, no studies report a correlation between age and SAN lesion rate and/or recovery after temporary damage. After our ROC analysis, we established a cutoff of our population at 72 years old, which is the age most representative of differences between the two groups.

The two questionnaire scores highlighted that there was a significant difference between the two groups in terms of daily life impairment, with younger patients having fewer problems in their daily lives. However, this difference was more evident when it came to the grade of recovery after six months. Both questionnaires showed that the younger group performed well at the 6-month measurement after surgery. This result could be related to the well-known lower capacity for neural regeneration in older patients [45,46].

Because of the limited sample size, there was no statistical significance in the difference between the two groups.

## 5. Conclusions

SAN lesions after neck dissection in HNC patients are one of the most frequent complications, causing worse quality of life in these patients. We demonstrated age as a risk factor that predicts either SAN lesion rate or recovery after surgery, which appeared to be better in younger patients. Sparing level IIB in older patients when oncologically possible could lower SAN lesion-related complications in older patients. Further studies and a larger sample are needed to confirm our results.

## Figures and Tables

**Figure 1 jpm-13-01082-f001:**
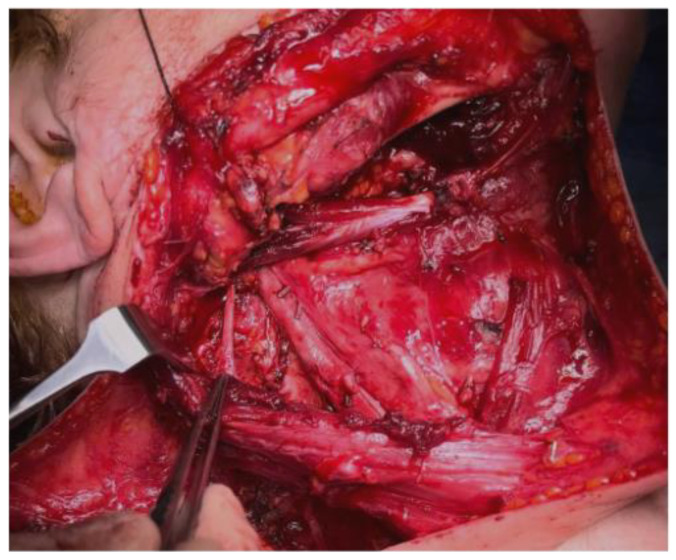
Anatomopathological view.

**Figure 2 jpm-13-01082-f002:**
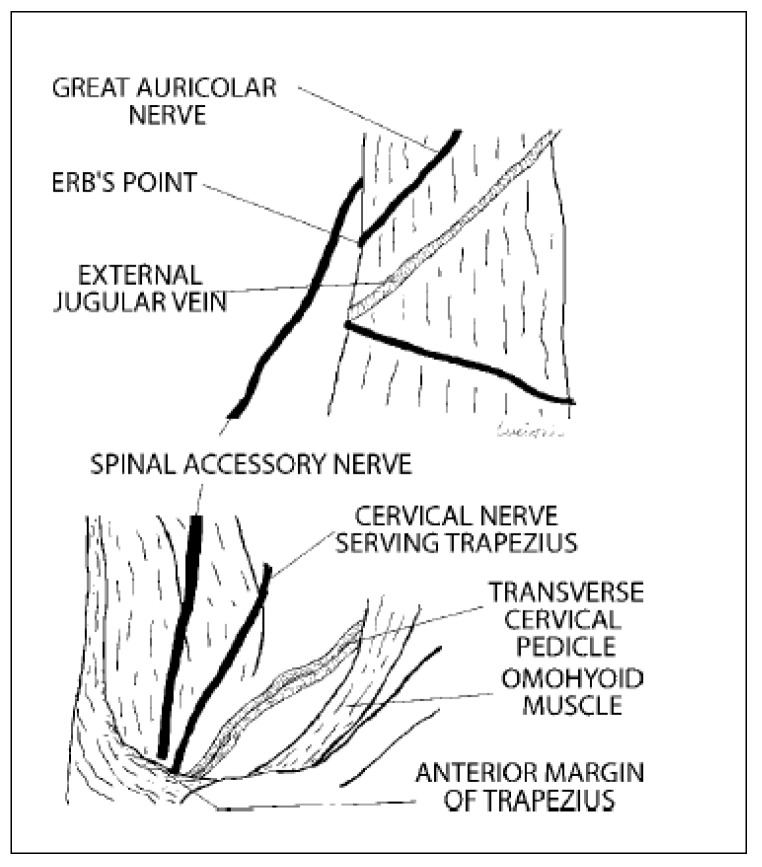
SAN’s schematic cervical course, image 6.4 from “Lucioni et al., *Practical guide to neck dissection*” [16].

**Figure 3 jpm-13-01082-f003:**
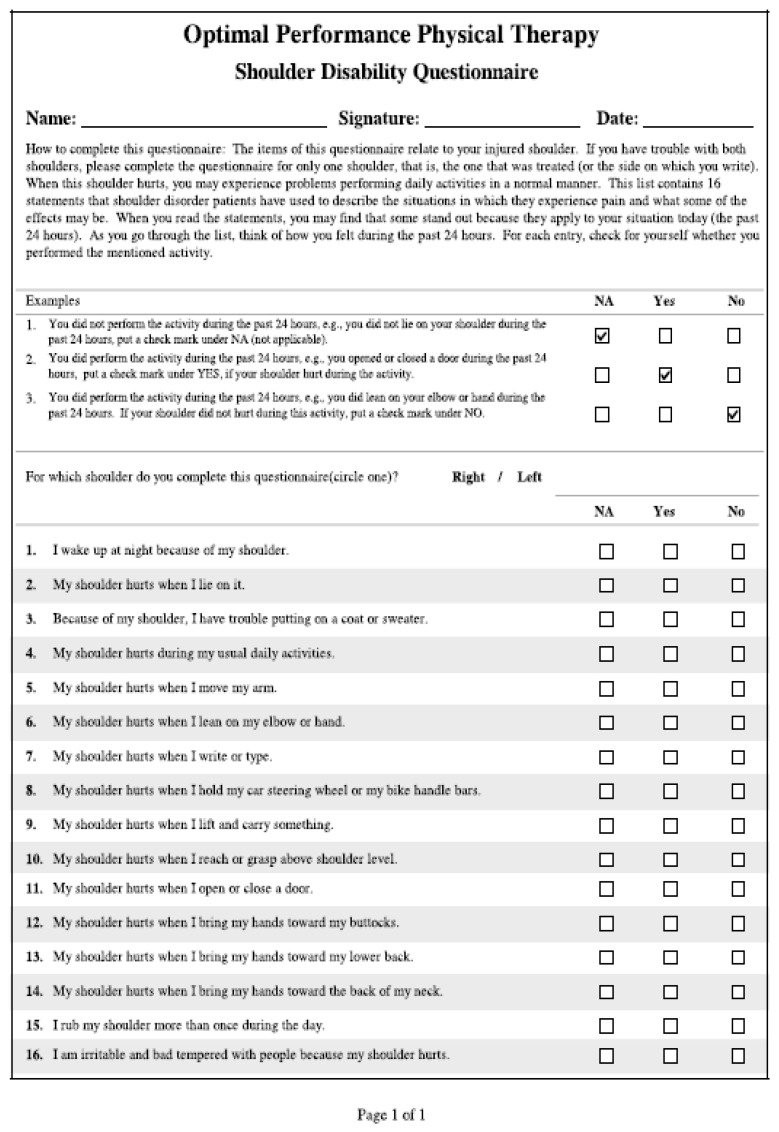
SDQ questionnaire.

**Figure 4 jpm-13-01082-f004:**
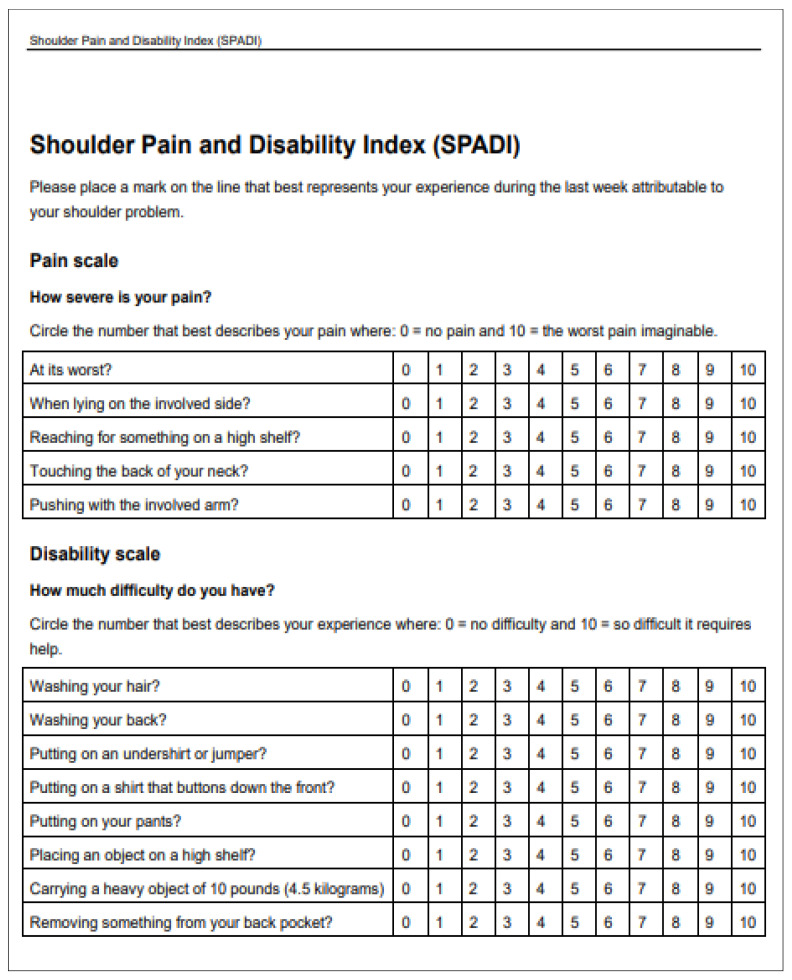
SPADI table.

**Figure 5 jpm-13-01082-f005:**
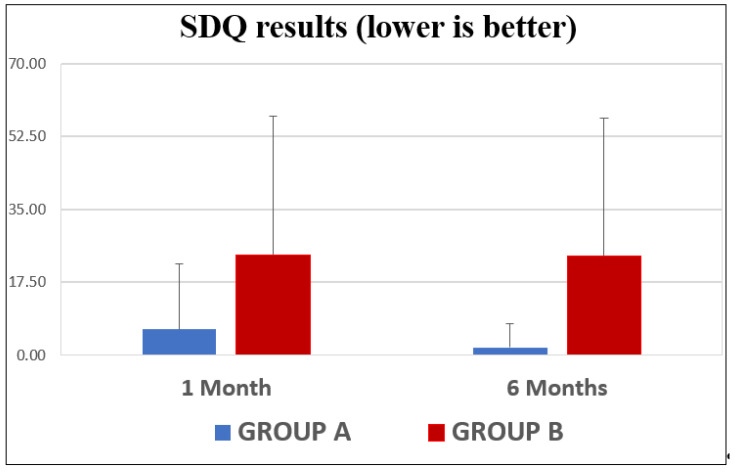
SDQ score differences between the two groups.

**Figure 6 jpm-13-01082-f006:**
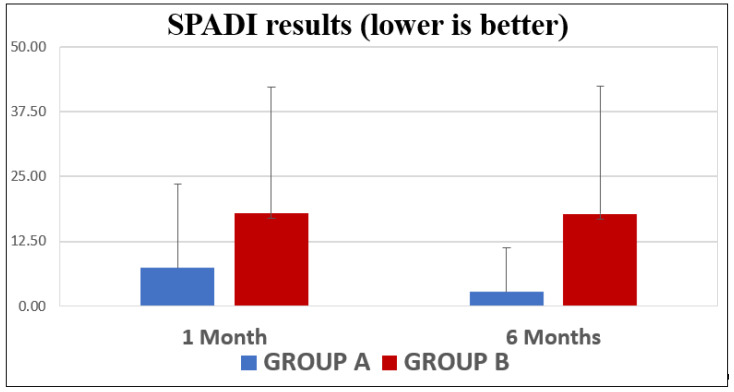
SPADI differences between the two age-based groups.

**Figure 7 jpm-13-01082-f007:**
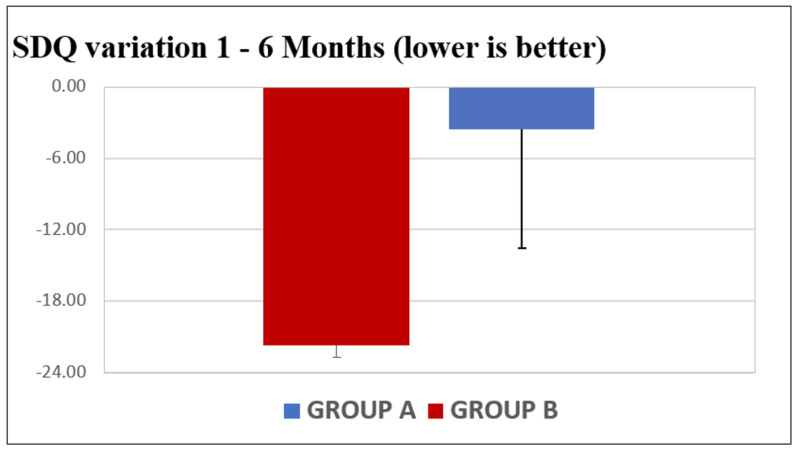
SDQ score variation between the two age-based groups.

**Figure 8 jpm-13-01082-f008:**
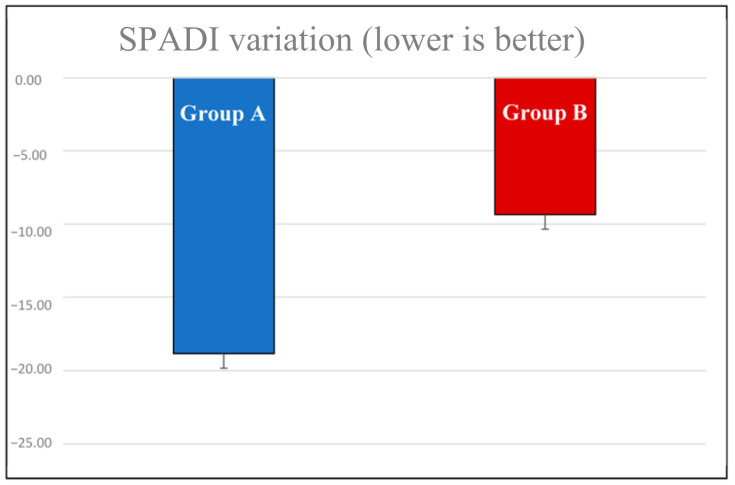
SPADI score variation between the two age-based groups.

## Data Availability

Data are available on request to the corresponding author.

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
