# Peer review of "Does Age Affect the Rate of Spinal Nerve Injury after Selective Neck Dissection? Age as a Prognostic Factor of Spinal Nerve Injury after Selective Neck Dissection"

_jpm, 2023, doi:10.3390/jpm13071082_

Round 1

Reviewer 1 Report

1.     Abstract: 

a.     you should explain the meaning of SPADI and SDQ (line 28);

b.     you wrote the study was carried out in patients with head and neck cancer; however, the included patients were suffering from only oral squamous cell carcinoma;

c.      in results section, you should report data about older age and increased risk of nerve injury;

2.     Typing errors in lines 39, 63, 65, 173, 209, 216, 217, 218, 222, 223…

3.     Figure 8: you should correct the title of the table;

4.     Methods: lines 146-147, I think the SDQ scale is wrong. Reading some articles, I found that it ranges from 0 “ best” to 100 “worst”;

5.     I do not think that either group had a significant subjective impairment: indeed, average scores are quite low (better);

6.     Methods:

a.     Who performed the surgeries? What surgical experience do they have? They are at least 2 surgeons with different skills since it’s a multicentric study (inter-individual variability);

b.     In my opinion, the same questionnaires had to be submitted to patients even in the pre op;

7.     Results: 

a.     In my opinion, a table with general patient data and the results of the questionnaires in order to have an overview would be useful;

b.     9. Did patients with impairment receive therapy (steroids? other?)? Or has any spontaneous recovery been evaluated?

c.      9. How many patients have undergone adjuvant RT? How were they distributed between the two groups? The result probably had to be reported separately. Moreover, the evaluation of the result was carried out at different times than patients without RT.

8.     Discussion: 

a.     You reported studies about the increased risk of spinal nerve injury during level IIB dissection; however, in your work, you do not mention it. So, did you dissect level IIB?  What percentage of patients? Did you find increased incidence of nerve injuries in patients undergoing dissection level IIB?

9.     Conclusions:

a.     In your work, you didn’t mention level IIB dissection; so, in my opinion, you can’t report it as a take home message in the conclusions;

10.  In the results section of abstract, you mentioned hypoglossal nerve but then you didn’t analyze its injury in the study;

11.  34/48 articles were published more than 10 years ago; you should add some more recent article.

Author Response

Dear Reviewers, 
We are very grateful devi your suggestions and have made all the requested changes, we hope we have answered all the points you requested. 

  1. Abstract:
  2. We explained briefly the meaning of SPADI and SDQ in the abstract“Two shoulder specific questionnaire (SPADI and SDQ) were administered to assess shoulder morbidity postoperativetly”
  3. It was a refuse, corrected in the lines 184-186 “We included a total of 78 patients; 75 patients who underwent levels I-II-III neck dissection for HNC (71 cancers were in the oral cavity, 2 on the nasal surface, 2 were located in salivary glands.), 2 patient of adenoid cystic carcinoma, and 1 of osteosarcoma”
  4. The typing errors were corrected
  5. We corrected the title of the table in: “SPADI variation (lower is better)”
  6. We corrected the SDQ scale “The SDQ (Figure 3) is a self-administered questionnaire which measures pain and movement impairment of the shoulder. It is composed of 16 items recalling daily life scenarios. All answers can be calculated as “Yes” = 1 point, “No” = 0 points or “Not applicable” = missing. The scale ranges from 0 = normal to 100 = worst disability”
  7. The questionnaire used investigate every-day tasks so even low scores can impact negatively onto the Quality of Life of patients.
  8. Methods:
  9. The procedures were performed by two surgeons that comes from the same surgical school in order to lower inter-individual variability.
  10. We rule out every patient that had any sort of pre-operatively shoulder impairment (exclusion criteria)
  11. Results:
    1. No
    2. Patients with impairment did not receive any sort of treatment, we evaluated the rate of spontaneous recovery
    3. Radioterapia
  12. Discussion:
    1. All SND performed included level IIB dissection: “Selective neck dissection of levels I-II (always including IIB)-III” line 143
  13. We clarified the role of SND in SAN injury in lines 91-96 “When it comes to level IIB dissection, to correctly isolate the IIB level surgeons must isolate the nerve from the fibro-fatty surrounding tissue, not only impairing its blood supply but also manipulating it with surgical instrument that can provoce stretching damages. Even during selective neck dissection, that exposes a limited part of the nerve compared to Functional Neck Dissection, SAN is vulnerable to iatrogenic injury”
  14. It was an error, we never considered hypoglossal nerve injuries.
  15. All article mentioned are the mile stone on the topic. We suggest to consider that them are still used for gold standard gudelins for the topic.

Reviewer 2 Report

The current study is aimed at determining if age affects the rate of spinal nerve injury after selective neck dissection treatment in patients of head and neck cancer. 

Please correct the following:

Title: The title can be rephrased to give a better understanding of the research. For example, "Impact of age on the rate of spinal nerve injury after selective neck dissection and head and neck cancer patients"

 Abstract: The abstract should explain the study in brief. The methods section can be improved by including the number of patients. The results section should include the findings of the research that is missing. It is expected to see how many patients and the average age of patients who had spinal nerve injury in this part of the abstract. Then it is easier to understand the conclusion, which, as per your paper, is that age does affect spinal nerve injury outcomes in head and neck cancer patients. Therefore, please work on the abstract section.

Lines 77, 78: Please explain the levels here, as it is an important part of the research.

Materials and Methods Section: Line 126 - Please mention exact dates here, e.g., 1st June 2020 to 30th June 2022.

In the "Methods section, describe the type of study. As per the abstract, your study type is a retrospective cohort study. Please mention this in your methods section. Also discuss the number of patients recruited. Discuss control group details. 

The research article says Institutional Review Board statement is not applicable.

Being a retrospective cohort study, I believe IRB statement is important.

Spelling mistakes:

Line 39: Pharunx-Pharynx

Line 63: oh-Of

Lines 40, 41: Rephrase the sentence.

Line 44: Underlines are under citation numbers, please correct as per journal instructions.

Lines 50, 51: Rephrase and edit this line.

Line 64: "patients" in place of "patient"

Line 65: "lymph nodes" in place of "lymph node."

Line 86: remove parenthesis; write "SAN" and not "(SAN)".

Line 92: Inconsistency in the citation system throughout the paper. Please correct it.

Line 105: "Shoulder" in place of "shoulders"

Line 140: Full form of CT

Lines 163, 165: Please correct the type of figure, Figure 3 SDQ Table—Figure 3 SDQ Questionnaire.

Line 168: 62,5 yo - write correctly: 62.5

Lines 190–192: Please arrange the document appropriately.

Lines 194-196: I am not sure why this is included in the manuscript. Please correct it

Figure 8 has some errors that need to be fixed. 

Lines 205-207: Please rephrase

Lines 215–224: The font style is changed. There are many errors through this paragraph. There is need to rephrase this paragraph.

Line 226: You can write " in the current literature" rather than "actual literature".

Lines 242, 245: QoL and ROC: Please write the full form.

Line 256: Little sample—you can write limited sample size.

Author Response

Dear Auditors, 
We are very grateful devi your suggestions and have made all the requested changes, we hope we have answered all the points you requested. 

  1. We propose a new title: “Age as a prognostic factor of Spinal Nerve Injuriry After Selec-tive Neck Dissection”
  2. All the mistakes of spelling are corrected following reviewer report
  3. Date was corrected
  4. IRB was bot necessary by our hospital board because all patient subscribe an informed consent before surgery for eventually use of anonymous data.

Round 2

Reviewer 1 Report

thanks for the corrections and explanations. 

abstract section: my only advice is to to explain the acronyms SPADI and SDQ.

Author Response

Dear Reviewer, 

We used in the abstract section the full denomination of the two questionnaires as you may see in line 30.

Best Regards 

Reviewer 2 Report

Thank you for sending us the revised manuscript.

Line 35: Ends in "and". Please correct.

Line 43: Rephrase the line, e.g., "They potentially metastatize to the lymph nodes in the neck."

Line 68: Remove also

Lines 75, 76: Surgery along with radiation and chemotherapy can be more appropriate here.

Line 100: The citation style should be uniform throughout the paper, please fix it.

Line 160: Use the word "analgesics", instead of "painkillers".

Author Response

Line 35: it was a typing error, it was deleted.

Line 43: We choose to rephrase in:”They tend to metastasize to the loco-regional lymph-nodal system.”

Line 68: we removed also

Dear Reviewer, 

Line 75; 76: We choose to rephase the sentence in “However multimodal treatment strategies such us radiation and chemotherapy are strictly indicated for advanced staged diseases in adjuction with surgery or in palliative interventions”

Line 100: We corrected the citation style.

Line 160: We replaced the term “painkillers” with “analgesics”

Best Regards